# Perfecting Bodies: Who Are the Disabled in Andrew Niccol's Gattaca?

**Chia Wei Fahn** 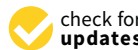

Department of Foreign Languages and Literature, National Sun Yet-sen University, Kaohsiung 80424, Taiwan;
celinefahn@gmail.com

**Abstract:** This paper will examine the impact of genetic technologies on the corporeal and economical aspects of human lives while emphasizing the ambiguity of disability under these subversive circumstances. In 2013, the world was introduced to CRISPR genetic editing technology, followed by the controversial announcement in 2018 from Chinese scientist He Jiankui, who claims to have genetically engineered twins that were born HIV-immune. The possible social outcome of genetic treatment leading to the alteration of human embryos to create physically and intellectually superior offspring, as well as its impact on the social treatment of disabled bodies, is clearly illustrated in Andrew Niccol's directive debut *Gattaca*. Here, I will discuss Niccol's utilization of disabled characters in interrogating the employment of disabled characters as a narrative vehicle to reflect upon social paradigms. I examine both the subversion and expansion of the social construct of disability in *Gattaca*'s narrative, emphasizing the film's portrayal of economic differences as a disabling factor in a world of augmentative technology.

**Keywords:** genetic engineering; He Jiankui; CRISPR; *Gattaca*; disability; posthumanism; prosthetics

---

Posthuman theories have emerged in recent years to encompass all areas of biocultural discussion that integrate the social and the scientific in critical thinking, with advocates applying interdisciplinary approaches to formulate a comprehensive understanding of culture and technology in the post-anthropocentric era. Discourses and representations of "the non-human, the inhuman, the antihuman, the inhumane and the posthuman proliferate and overlap in our globalized, technologically mediated societies" [1] (p. 2). This interdisciplinary discourse between human, social, natural, cognitive and bio- or life sciences came to focus on an intersection between posthumanism and disability studies, with an emphasis on the unique role of science fiction in shaping contemporary technoscapes [2]. Science fiction calls attention to pertinent issues that surface as the human body and advanced technologies continuously merge, with Kathryn Allan describing SF as "a genre that criticizes the politics and ideologies of the current day, as its writers imagine the possibilities of future worlds" [2] (p. 1). David L. Wilson and Zack Bowen interrogate ethical and moral concerns in relation to scientific advance; they believe that literature works serve as a cautionary narrative against the effects of technoculture, with SF acting as "both a warning of the adverse possibilities of taking certain paths and a deterrent to the scientific hubris that could put us on those paths" [3] (p. 201). Through science fiction, future possibilities are explored from the approaches of "experimentations with the boundaries of perfectibility of the body, in moral panic about the disruption of centuries-old beliefs about human 'nature' or in exploitative and profit-minded pursuit of genetic and neural capital," all of which pose a direct challenge to the physical boundaries between normality, biodiversity, and disability [1] (p. 2).

Biotechnology asserts with "unshakable certainty the almost boundless capacity of humans to pursue their individual and collective perfectibility" [1] (p. 13). The two largest developing areas

of prosthetic biotechnologies that "perfect" the contemporary body lie in the research of physical augmentation through genomic research and bionic engineering. Apart from the technological developments in progressive bionics, our society has witnessed an astronomical advance in genomic study under the universal goal of identifying and researching cures for hereditary disorders. Genetic research promises to cure diseases, prevent disability, induce longevity, and improve the human condition in general, yet its application "requires serious speculation, and the vocabulary of speculation should not be what genetic science can do, but what it can do to, or for, society" [4] (p. 9). These speculations "raise deep anxieties both about the moral status of the human and express the political desire to resist commercially owned and profit-minded abuses of the new genetic know-how" [1] (p. 40). After two centuries of pursuit of 'perfecting' the human species, the posthuman condition "introduces a qualitative shift in our thinking about what exactly is the basic unit of common reference for our species, our polity and our relationship to the other inhabitants of this planet" [1] (p. 2). This has urged theorists to explore the potential of genetic engineering, as well as expounding upon issues pertaining to the inevitable impact we face as individuals and as a collective society. Technological impact on both the "normal" way of life and normalized bodies are popular areas of debate, yet the reflexivity of the disabled body in reaction to socio-economical change, and a subsequent subversion of both the definition and corporeal experiences of disability, have been largely overlooked. This article will examine the impact of genetic technologies on the corporeal and economical aspects of human lives while emphasizing the ambiguity of disability under these subversive circumstances.

Over the course of the twentieth century, geneticists first succeeded in isolating DNA and have since then made immense progress toward mapping the human genome, opening the field of genomics and corresponding studies in how our physical functions are programed through molecular interaction. We now know DNA to be strands of amino acid in a constant state of change and response to chemical reactions and environmental stimulants, though genomic research continues to decipher the delicately balanced interactions and reactions between each gene and its corresponding physical expression. Not only is our corporeal existence maintained by an intricate, internal network of neurological and chemical commands; study in genealogy has firmly established the body as an open site that never ceases to shift in action and reaction as the proteins, hormones and enzymes in our cells respond to constant external stimulus. Genomic knowledge has risen to such commercial importance that joint ventures in both the private and public sector race to produce the latest breakthrough. The Human Genome Project is one of these expeditions dedicated to genomic research in developing new ways to treat, cure, or even prevent the thousands of diseases that afflict humankind. Donna Haraway articulates the practice of interpreting through genetic code by stating:

> In modern biologies: the translation of the world into a problem in coding can be illustrated by molecular genetics, ecology, sociobiological evolutionary theory and immunobiology. The organism has been translated into problems of genetic coding and read-out. Biotechnology, and writing technology, informs research broadly . . . . Immunobiology and associated medical practices are rich exemplars of the privilege of coding and recognition systems as objects of knowledge, as constructions of bodily reality for us. Biology here is a kind of cryptography. Research is necessarily a kind of intelligence activity [5] (pp. 162–163).

This deconstructive treatment of the human body furthers research in understanding the chain-reactions directed by our DNA, and identifies the genetic sequence as an innate, written manual that "programs" the body from its conception.

In addition to genomic studies, progress in medical imagining such as CT and MRI scans, pharmacology, and neurobiology have also contributed to treating the body as a programmed database [6] (pp. 84–85). Pramod K. Nayar describes this posthuman deconstruction as a "databasing of the human population," which, combined with the rise of systems biology, drastically alters the perception of the body as a coherent, unified, well-bounded entity [6] (p. 84). Nayar contends that systems biology and genealogy exposed the human body as:

Essentially a collection of links in which even the DNA is one of several components with little more than an average role to play. What is interesting is how such a view intersects with the idea of life reducible to its chemical processes and components. Biology is interpreted as the manifestation of a set of chemical processes and codes. It is also seen as a consequence of the networks of chemical, neurological and other processes. While the organism is reduced to, or mapped as, a genome, the genome itself must represent life itself ... this view of life rejects the earlier idea of a bounded self/body and an invisible life force [6] (p. 84).

Under this premise, our bodies are understood as the embodiment of multiple forms of data. James C. Wilson explains the concept of "reading" the human body as "a molecular language where the 'letters' are bases, the 'words' are genes and the 'book' is the complete genome" [7] (p. 68). Therefore "genetics become textuality, and the human genome becomes the 'Book of Life'" [7] (p. 68). The basics of genetic engineering fall in line with viewing the human body's "Book of Life" as a raw text, and biotechnological treatments an act of revision [7] (pp. 67–68). Research in genetic engineering strives to edit the "defects of faulty genes," to "correct" errors in our genetic material as one would "correct" an "incorrect" text [7] (p. 69). Consequently, genetic editing treats the human body as "an assemblage of parts, conceived of in terms of a machine that can be fully understood, operated, repaired, and redesigned" [8] (p. 260).

James Wilson's work in discussing the pros and cons of geno-determinism deconstructs the containment of individual bodies and expounds upon the importance of understanding the human body as a recipient to external stimulus. The reading of genomic data does not predetermine individual lives, he argues, as the genome is "always in the process of becoming. Genomes are dynamic, constantly evolving over time, shaped by both internal and external factors" [7] (p. 69). Nayar observes that "the genetic constitution of an organism is not ... bounded by the organism's body: it is in linkage with other organisms and the environment .... The subjectivity of a human lies not within the interior but within the exchange and linkage the interior has with the exterior world" [6] (p. 98). Our genomes are a sequence of actions and reactions between the internal and external environments as well as physical changes. With each living millisecond, countless cells in our body die and fresh cells take their place, forming a new physical space dramatically different from its predecessor that results in an ever-changing state of chemical and physical expressions [7] (pp. 70–72). In the process of genetic editing, certain parts of the subject's cells are cut and removed from the body, whereas enhancive extensions are added to create augmented alterations.

In 2013, research developments on the successful removal of a blood disorder gene using new CRISPR technology were made public, leading to great excitement in the field. CPISPR (clustered regularly interspaced short palindromic repeats) employs enzymes that target specific stretches of genetic code and cut through the DNA strand. The cut DNA strand is then repaired by inserting a new stretch of genetic code, essentially achieving a find-and-replace functionality in the human genome. Genetic editing is essentially a form of prosthetic technology in both its execution and application [9] (pp. 75–76). The newly inserted genetic code is a designed prosthesis that serves as a prosthetic enhancement to the cut strand of DNA, and redirects correlating genomic expressions [10] (pp. 820–821). Optogeneticist Feng Zhang and his team at the MIT Broad Institute presented the first successful demonstration of Cas9-based genome editing in human cells, though uncontrollable mutations to both ends of the snipped DNA led the team to pronounce human application unfeasible at the time [11] (p. 12). Zhang and his team continued their experiments in the implementation of CRISPR technology, developing the editing tool to adapt to the current CRISPR Cpf1 version, which allows for a more accurate and efficient integration of DNA inserts. CRISPR Cpf1 operates on the same fundamental system as its predecessor, but instead of cutting through the entire strand of DNA and creating mutable breaks, Cpf1 is able to target and convert at the base point in order to treat the corresponding disease [11] (p. 15). The Zhang Lab's success revolutionized geno-determinism studies, proving that the human genome can in fact be edited to eliminate inherent conditions and allow possible enhancement in the future.

Zhang's team is but one in the many around the world that push toward a working model of genetic modification. In 2017, CRISPR technology proved to be pivotal in removing the HIV-1 genome and correcting inherited cardiac diseases in human embryos grown in vitro [12–14]. Chinese biophysics researcher He Jiankui announced his team's success in engineering the world's first genetically altered babies in 2018, two days before the Second International Summit on Human Genome Editing in Hong Kong [15] (p. 33). He followed by explaining the process at the Hong Kong Summit, claiming that his team had edited embryos created from HIV-positive sperm paired with HIV-negative eggs by using (CRISPR)-Cas9 technique. He Jiankui and his team targeted and disabled the CCR5 gene, which controls the protein doorway that allows HIV to enter human cells [15] (p. 33). The procedure resulted in a pair of twins immune to the HIV virus and caused the existing controversy surrounding bioengineering and genetic research to climax in an international burst of ethical outcry. He Jiankui was denounced comprehensively for his blatant disregard of bioethical boundaries; however, the world's first genetically altered babies are an irrevocable part of contemporary reality. The case reveals the discrepancy between technological advancement and corresponding ethical regulations that highlights an urgent need to reinstate the boundaries between eugenic enhancement and providing cure; an issue that demands our immediate attention [15] (p. 36).

Lennard Davis traced the origins of eugenic thought and engineering of physical alterations in "Constructing Normalcy: The Bell Curve, the Novel, and the Invention of the Disabled Body in the Nineteenth Century" to the rise of statistical study [16]. Davis' article drew attention to the work of French statistician Adolphe Quetelet's application of the astronomical "law of error" in calculating the average of man [16] (p. 4). Quetelet's average man is both a physical and moral construct, replacing the ideal body with the ideal of normal [16] (p. 5). Quetelet believed that under an emphasis and pursuit of statistic averages, all physical deviation will gradually diminish alongside the body's "defects and monstrosities," therefore allowing humanity to enter a "utopia of the norm associated with progress" and pursuit of a "perfectible human species" [16] (pp. 5–7). Quetelet's views on the perfectible human body came to be broadly utilized in nineteenth century philosophy, most notably in the field of medical study which ultimately contributed to the rise of eugenics [16] (p. 7). The English statistician and social scientist Francis Galton coined the word "eugenics," a term referring to his idea of selective breeding in order to enhance "desirable" and eliminate "undesirable" traits [16] (p. 8). Though Nazism discredited the eugenic movement after the second World War, eugenics reemerged as the new field of genetic engineering with the advent of biotechnology in the 1970s. Disability studies scholars posit that this eugenic selectivity remains at the core of genetic engineering, with Snyder and Mitchell arguing that "the updated eugenics of the present day, called genetics, examines conditions in bodies that are classed as . . . 'tragic,' 'coding errors,' 'suffering,' 'unhealthy,' 'deviant,' 'faulty,' and 'abnormal'" [17] (p. 19). Davis refers to genetic engineering as a process that "carries a weighty signification;" these treatments seek to better the disabled body yet also view disability as a "defective" state that must be "altered" [9] (p. 102).

Tobin Siebers believes that "curing" disability serves to maintain and create "quality human beings;" an unrestrained pursuit will inevitably further the disability stigma [18] (p. 4). In the introduction to *Disability in Science Fiction: Representations of Technology as Cure*, Kathyrn Allan writes:

> The medical characterization of the disabled body as requiring cure . . . has become part of our larger cultural construction of disability . . . . The ideology of the perfect body—and our ability to make imperfect bodies perfect . . . is woven throughout our various social discourses, and the onus to be a perfect body rests on both the abled and disabled alike [2] (p. 9).

Science and technology offer a "solution" for disability, a transcendent way of life that far surpass rehabilitation or even the normalized body [2,19,20]. Technological cures "enable . . . return to an able-bodied functionality, or even to gain an enhanced functionality and become stronger, faster, or more responsive" [21] (p. 1040). The narratives of technology as a cure, argues Mary Seaman, "react against a techno-posthuman world in which the key components of the human . . . are viewed as

inherently flawed, with science as the rescuer of the human from its mortal self . . . the human becomes an assemblage of parts, conceived of in terms of a machine that can be . . . repaired, and redesigned" [8] (p. 260). There has yet to be an open marketing of genome enhancement in our contemporary era under the ethical safeguards of vehement eugenics opposition from critics; the advertisement of the genomic progress is carefully clothed under the shroud of cures for cancer or the next designer drug. However, bioethicists such as Jeremy Rifkin also caution against the efforts to control and alter our genetic makeup:

> If diabetes, sickle-cell anemia, and cancer are to be prevented by altering the genetic makeup of individuals, why not proceed to other less serious 'disorders:' myopia, color blindness, dyslexia, obesity, left-handedness? Indeed, what is to preclude a society from deciding that a certain skin color is a disorder? In the end, why would we ever say no to any alteration of the genetic code that might enhance the well-being of our offspring? [22] (p. 140).

Regardless, genomic alteration is increasingly recognized as an up-and-coming market with genetic engineering targeting hereditary disease on the one hand and furthering pharmaceutical cure on the other.

Once fully established, genetic treatments will undoubtedly provide further assistance toward the disabled, though their development also offers an opportunity for physical alteration and the possibility of creating a perfected, enhanced body. Genetic engineering promises us a future that is free from hereditary disabilities, but where do we draw the line between cure, care, and eugenic discrimination? In 1997, Andrew Niccol introduced his original screenplay and film, *Gattaca*, as a direct inquiry to the future of genetic engineering that is particularly relevant to the controversial developments in contemporary society. In the decades that followed its premiere, the *Gattaca* narrative has been evoked time and time again as a precautionary tale when advances in genetic research pronounced this "not so distant future" taking one step closer to reality [23]. Michael Bérubé identified *Gattaca* as "a film about civil rights. It is, I might say, one of the few science fiction films that is centrally concerned with discrimination" [24] (p. 207). Ron Von Burg notes that Niccol's portrayal of geno-research "shifts genetic science and genetic information from an epistemic issue to an ethical and legal one" [4] (p. 13). David Kirby "warns of the problems that arise if we believe that humans are nothing more than their genes" [25] (p. 198). In the film's opening sequence, unknown particles float and swirl through the air in a flurry to land upon a sterile surface. The particles are hair follicles, fingernail clippings, whiskers and flakes of skin, magnified to resemble tree trunks, elephant tusks, firewood, and slate. As the camera zooms out to enlarge the scene, viewers see a naked man vigorously scrubbing away at his body in a steel incinerator. The man exits and ignites the incinerator with a resounding thud, symbolic of the weight and power that our bodies' fragments have come to wield in modern society. With one scene, Niccol illustrates the posthumanist system in which the human body has been deconstructed into genomic matter by global networks of control and commodification. *Gattaca* is a bioethical text that addresses the issues associated with genetic therapy and the new eugenics by questioning the ideology of genetically engineering in order to warn the contemporary viewer against a fervor to "correct God's mistakes" [22] (p. xii).

Niccol visualizes a world where genomic technology successfully identifies and cures hereditary diseases. Although genetic alterations were not made mandatory, parents who have the means to submit their DNA to a screening process are then encouraged to select an optimal genetic paring to guarantee children their best genetic attributes. Individual life expectancies and physical health statistics are read minutes from birth, where one's "destiny was mapped out" through DNA testing [23] (p. 12). At first glance, *Gattaca* presents an optimal image of the future with its sleek urban landscape, advanced gadgets and attractive occupants. Upon close inspection, however, the men and women portrayed as the social elite have a suppressed air of manufactured uniformity in spite of an emphasized racial and gender inclusion. All wore "a certain arrogance, a cool detachment" with "similarly unconventionally-cut suits, short coiffed hair and robust tans," exuding the glow of perfect physical health [23] (p. 5). The camera follows the tall, blond man seen exfoliating earlier through the gates of the exclusive Gattaca

Aerospace Corporation, where blood samples were taken from all employees to confirm entry. A screen flashes with a photo identification of "Jerome Morrow" and the word "valid." Introduced as member of the social elite, Jerome describes his world as made to order; by eradicating any "weak" link in genetic pairings, this eugenic society only preserves the best in genetic heredity and ostracizes those who were not submitted to the same process. The film provides a first-person narration by "Jerome Morrow, Navigator First Class," whose "selection . . . was virtually guaranteed at birth. He is blessed with all the physical and intellectual gifts . . . a genetic quotient second to none" [23] (p. 9). In a detached voice, Jerome states that his success was unremarkable, the only surprise was his identity; he was not Jerome Morrow.

The *Gattaca* universe fully utilizes genetic knowledge to exert control over the individual and governs by compiling a genomic database at each childbirth. Individual identities are reduced to genetic code. Niccol puts the social effects of genetic engineering into stark reality; the dominant class in its social hierarchy are genetically screened, therefore eliminating any chances for those who do not undergo alteration. "Jerome" is soon revealed to be naturally conceived Vincent Freeman, ruled at birth as intellectually and physically inadequate. His genetic reading revealed the possibility of heart diseases, violence, psychotic inclinations and "prone to learning disabilities" [23] (p. 12). As an unaltered child, Vincent is constantly discouraged from any form of physical or mental activity that may stress his "defective" genes, his "flaws, predispositions and susceptibilities—most untreatable to this day. Only minutes old, the date and cause of my death was already known. . . . wherever I went, my genetic prophecy preceded me" [23] (pp. 12–13). The film shows a young Vincent being led away from classrooms where he was constantly snubbed and put in rooms with children who have visible disabilities. Vincent was deprived of a formal education as his "genetic scarlet letter continued to follow [him] from school to school" [23] (p. 12). Vincent's counterpart is Eugene, his wheelchair-bound roommate who possesses a superior genetic sequence yet was unable to fulfill his highly anticipated potential. Eugene attempts suicide only to wake a quadriplegic; he is forced to peddle body matter to maintain a luxurious standard of living. To secure a position within the elite Gattaca corporation, Vincent assumed Eugene's geno-identity and subsequently rose rapidly in corporeal ranks. Together the two men formed a "borrowed ladder," a "member of a relatively new and particularly detested segment of society—one of those who refuse to accept [genetic] destiny . . . . most commonly known as a 'de-gene-rate'" [23] (p. 37).

Apart from a critique of genomic control and exclusivity, *Gattaca* is also a film about disability. Disability studies deals with challenges in understanding disability and the oppression of physical and social norms that connect the political, economic, and biological; all of which converge at the disabled body as a prosthetic site for interdisciplinary interrogation. The disabled body is "an interface where intersecting material and symbolic forces converge . . . therefore, an opportunity to think through values, ethics and politics" [26] (p. 636). David Wills views this interface as a prosthesis that is a mediation between the literary and the body [27] (p. 266). David Mitchell and Sharon Snyder discuss the literary representation of disabled bodies as a prosthetic interface in *Narrative Prosthesis: Disability and the Dependencies of Discourse*, to read disability and social rhetorics within the domain of narratives. Mitchell and Snyder demonstrate the employment of disabled characters that are utilized in literature as a narrative prosthesis, allowing literary works to comment upon and challenge the homogeneity of society by revealing its tensions and paradox [28] (p. 47). They argue that the disabled have long been utilized in literature as "a narrative prosthesis; their functions are restricted to either serving as metaphorical warning or a representation of morality and mortality" [28] (p. 47). Disabled characters either represent "presumed malignancy or excessive fragility," thus furthering the negative misrepresentation of disabled persons [28] (p. 21). In "Screening Stereotypes: Images of Disabled People in Television and Motion Pictures," Paul Longmore identified the three main disabled stereotypes in modern media: "disability is a punishment for evil; disabled people are embittered by their 'fate'; disabled people resent the nondisabled and would, if they could, destroy them" [29] (p. 67). Both arguments saw literary depictions as an influential force in shaping derogatory attitudes

about people with disabilities, reaffirming how negativity is reinforced by restrictive paradigms, tropes and plots. Mitchell and Snyder's work pursues the subject of disability from the direction of "character-making trope in the writer and filmmaker's arsenal, as a social category of deviance, as a symbolic vehicle for meaning-making and cultural critique, and as an option in the narrative negotiation of disabled subjectivity" [28] (p. 1). With this perspective, my following analysis will examine disability in *Gattaca* as a narrative prosthesis in order to discuss a crucial question: in a world where genomic treatment is the ultimate cure, who are the disabled?

Niccol's vision of our future in genetic engineering encapsulates the eugenic ideology behind the bid to create genomic cures, yet disability remains an inescapable part of physical existence in the *Gattaca* universe. Niccol creates a foil through the relationship between Eugene and Vincent; Vincent is the able-bodied goal driven male, whereas Eugene is the emasculated invalid reliant on the care of others. As time goes by, Vincent is able to overcome his genetic limitations and deliver daily the work performance required of the genetically altered elite. His progress is seen as a reaffirmation of faith in the potential of the able-bodied working-class male, allowing the viewer to find reassurance in their own normality. In contrast, Eugene relies on Vincent to provide an income that fits his needs while he stays in their shared apartment drinking his days away. Kathleen Ellis vehemently criticizes the film's contrast of the physically impaired Eugene as a supporting character to Vincent's able-bodied success, stating that *Gattaca* "employs stereotypes of disability as a narrative tool to reassure audiences of their 'normality.' In the same way, disabled audiences are reminded that they are not valued members of their society" [30] (p. 112). Eugene's role as the atypical disabled character is in many ways stigmatizing. *Gattaca* shows Eugene's experiences of disability in intimate detail, a stark contrast to his sterile surroundings. Little by little, he descends into a "clichéd story of a monstrously tragic cripple" unable to face his failure and disfigurement [31] (p. 144). Eugene remains hidden from the public eye and grows increasingly pale in comparison to Vincent's glowing success, while wallowing in self-pity and depression. In one particularly distinctive scene, Vincent stands tall and arrogant on a helix-shaped staircase connecting the two floors of the loft apartment the two men share, looking down at Eugene's contorted body hunched over his wheelchair, "his paralyzed legs a particular dead weight" [23] (p. 3). Eugene, the engineered genetic wonder, was weak and maimed, while Vincent, the strong, able-bodied everyman, rises above his humble origins and stands victorious. Eugene becomes invisible to conceal Vincent's dependency and is gradually dehumanized as Vincent's "homo-affectionate roommate, then his housewife, then his child, then his pet, and finally his burnt offering" [31] (p. 149). In the end, Eugene crawls into their apartment's incinerator and dies in a fiery inferno, having served out his narrative function as the "monster's fallen body as stepping stone" [31] (p. 150).

Christopher Krentz describes *Gattaca* as a film that attempts to erase disability by "having their disabled characters commit suicide or go to great lengths to pass successfully as normal" [32] (p. 199). However, I argue that *Gattaca* is a film that presents disability as a point of resistance to reinforce the importance of individual diversity as opposed to conformity. Though Eugene's representation of the disabled subject initially provides an easy target to criticize the treatment of disability in this film, his character trajectory as an engineered marvel's fall from grace should not only be read as a supporting prop to facilitate able-bodied Vincent's journey to the stars. Mark Jeffreys questions Eugene's exile as an entrenched elite, "particularly when the injury does not affect the yardstick used to measure status in the first place, in this case his DNA" [31] (p. 148). When Eugene and Vincent first met, Vincent received a leg extension surgery in order to reach Eugene's full height before the accident. It is hard to believe that this world lacks the medical technology to treat Eugene's paralysis when genomic editing is the norm and reconnecting neurons after physical enhancements such as leg extensions are easily perfected. Eugene's confinement to his wheelchair therefore signifies a choice; his disability is not a question of if he can be treated or not, but whether he wishes to remain as he is. Eugene's tragedy is not his physical impairment, but his genetic superiority; he was not crippled by the inability to walk but by the crippling expectations of his social standing. Because his success is guaranteed, Eugene's best efforts are taken for granted, and anything that falls short is unacceptable. With all the physical

means to maximize achievement, Eugene is still unable to meet the expectations of a highly anticipated future. Once a college swimming champion, he came in second place in a race only to be disowned by his family, who had invested heavily to ensure his genetic material was unparalleled. Eugene's identity revolved around his engineered birth, his full name—Jerome Eugene Morrow—is a wordplay on "genome eugenics" and a constant reminder of the expectations weighing on his shoulders. The film depicts Eugene as constantly surrounded by bags of his blood, urine, snippets of hair and skin; Eugene the individual is engulfed and erased by Jerome, the genetically manufactured perfect body. Paradoxically, the only way to ascertain his individuality is to remain physically impaired. Eugene stands out as the only visibly disabled character in a world where individual bodies must conform and assimilate. Disability is thus an unstable category, transcending the boundaries that remain between body and diversity through which we can understand exclusion and resistance.

Eugene's disability is a rebellious statement of diversity. With outstanding genes, he was an unremarkable part of a homogenized society; he perversely only stands out after sustaining physical impairment. However, Eugene does not truly represent the disabled in the Gattacan society despite being visibly impaired. The interpretations of Eugene's character as reinforcing the disability stigma serve as a valid critique, yet disability in *Gattaca* should not be read as conventional, but as an unstable category. Amongst the genetically enhanced elite, the unenhanced Vincent is categorized as disabled whereas Eugene represents the social norm. Although Vincent eventually becomes a star navigator, he only does so by passing as able, as "Valid." Together, Vincent and Eugene constructed the shared identity of Jerome Morrow; Eugene provides the requisite body matter for the many checkpoints and security entrances that safeguard the Gattaca institute's genetic integrity while Vincent alters his physical appearance to pass as Eugene's doppelgänger. Vincent wears tiny packets of Eugene's blood under synthetic skin, and walks around with a bag of urine strapped to his leg. He leaves Eugene's hair follicles and nail clippings at his work station each day while meticulously wiping away all traces of his own existence. "Jerome Morrow" becomes a prosthetic identity for both men to pass as able in a super-able world. Jean Baudrillard cautions against the impact of genetic technology, seeing it as "the point when prostheses are introduced at a deeper level, when they are so completely internalized that they infiltrate the anonymous and the micro molecular core of the body ... burning out all subsequent symbolic circuits in such a way that everybody is now nothing but an invariant reproduction of the prosthesis" [33] (p. 119). This is clearly seen in Eugene and Vincent's relationship; both are consumed by maintaining the prosthetic existence of Jerome Morrow. Eugene's days are spent preserving bits and pieces of his "valid" body while Vincent obsessively scrubs away all vestiges of his in-valid self. The choice to become a "borrowed ladder" and remain confined to a wheelchair is therefore Eugene's act of resistance; by cloaking Vincent with Jerome Morrow's genetic identity, Eugene allows Vincent to become visible in a world where the genetically unaltered are invisible. Through the two men's relationship, Niccol commands the viewer's attention toward the unstable boundaries of physical ableism, inviting further discussion of the social model of disability.

The social model of disability identifies systemic barriers and social exclusion as the main contributing factors in disabling people [18,34,35]. *Gattaca* depicts a new social model of disability, displacing the role of current physical norms in a "cascading series of replacements reminiscent of Derrida's ideas of deferral or deferring—one replacement replacing another" [9] (p. 78). A social model of disability is employed here to show disability as a socially constructed identity. Niccol brings to public attention a provocative discourse on the issues that arise with the blind acceptance of genetic determinism and disabling the able-bodied. The disabled are genetically stigmatized, offspring of "those parents who, for moral or, more likely economic reasons, refrain from tampering with their offspring's genetic makeup or who fail to abort a deprived fetus" that "condemn their children to a life of routine discrimination" [23] (p. 12). What began as a means to rid society of inheritable diseases has become a way to "design your offspring—the line between health and enhancement blurred forever" [23] (p. 36). The possibility of genetic manipulation extends the scope of ableism, and things we now accept as "normal", such as myopia, obesity, and addiction, all present opportunities

to perfect the species. Thinking back to the day of his birth, Vincent nostalgically remarks that "not so long ago, I would have been considered a perfectly normal, healthy baby. Ten fingers, ten toes. That was all that used to matter" [23] (p. 13). In a world where genetic screening is prerequisite to procreating, however, he is categorized as "[o]fficially [as] 'In-Valids,' also known as 'godchildren' . . . 'deficients', 'defectives', 'geno-junk' . . . 'the fucked-up people'" [23] (p. 13). Average physical and intellectual normality is no longer the social ideal with the widespread administration of genomic treatments, causing normal and able bodies by today's definition to gradually become part of the disabled and disenfranchised. Genetic editing extends the boundaries of ableist ideology to view all unaltered bodies as disabled, potentially opening an oppressive social caste that is more exclusive than the present disability dichotomy.

Johnson Cheu's essay, "De-gene-erates, Replicants and Other Aliens: (Re)defining Disability in Futuristic Film", addresses future shifts in the social model of disability, arguing "if disability, as a social construction, exists on more than a theoretical plane, disability should be present as a social stigma in the future . . . which occupy a social stigma of being unfit, sub-human, inferior, that shows the very existence of disability as a social construction" [36] (p. 202). Disability in *Gattaca* remains a fixed category which humanity cannot transcend, "an atavism representing the remainder of normal at the end of normal" [9] (p. 14). Regarding genomics' impact on the social political as well as economic class structures, *Gattaca* subverts the belief that genetic engineering betters lives for those who live with disabilities with an emphasis on the dissonance between promised potential and social reality. Unscreened births are categorized as "the 'healthy ill'. They don't actually have anything yet—they may never. But since few of the pre-conditions can be cured or reversed, it is easier to treat them as if they were already sick" [23] (p. 37). Genetic editing led to an amplified expression of ableism in human society, by decreeing those we deem normal by today's standards the healthy disabled, "an unfinished and inherently malfunctioning organism" [8] (p. 260). Niccol predicts a future when our entire population is presumed to be in need of a cure. An "error free" biological profile becomes the sole determinant of ableism, an eradication of the "norm" we are accustomed to leads to further polarization of social and economic classes. Vincent's self-perception is severely influenced by those surrounding him, a continuous undermining of his efforts in life while "from an early age I came to think of myself as others thought of me—chronically ill. Every skinned knee and runny nose treated as if it were life-threatening" [23] (p. 13). Niccol articulates genetic technology's impact on society, examining how fundamentals in labor, class, and disability are redefined through the rise of genomics to form "a relation of power and substitution continuing from the social to the genetic" [9] (p. 81). For better or worse, genetic manipulation pushes the boundaries of what is previously considered as the human norm; all bodies are seen as "limited, defective and in need of constant improvement beyond species-typical boundaries" [37] (p. 254).

These deliberations on the relationship between disability, corporeality and materiality cement the posthuman approach as crucial in delineating humanity's collective experience of change by focusing on alterations of the way individual bodies interact with technological advances. Mariah Crilley posits that "science studies and new materialism offer a language and an ethics with which to account for a nuanced and dynamic materiality for the inseparability of body and self, of disease or disability and self, even microbe or gene and self" [38] (p. 306). Goodley, Lawthom and Cole believe the disabled subject "demands and affirms interdependent connections with other humans, technologies, non-human entities, communication streams and people and non-peopled networks;" disability is "the quintessential posthuman condition: because it calls for new ontologies, ways of relating, living and dying" [39] (p. 352). The question is, what ontological, ethical and political issues are raised by the access to technological treatments? Will the power to genetically alter, bionically substitute and otherwise hybridize the materiality of physicality ultimately decenter western definitions of human corporeality and individual agency? Or, contrarily, will biotechnology subject bodies to even more severe regimes of policing so that the "human capital" becomes a new and more flexible form of social caste?

Francis Fukuyama believes that a commercialization of geno-technology will lead to "the makings not just of a moral dilemma but of a full-scale class war" [40] (p. 16). After all, genetic engineering promises freedom from hereditary diseases, with the added bonuses of possible enhancements such as cognitive capabilities, athletic prowess, and an extended life expectancy, just to name a few. And why stop there? Vincent quips that "[e]yes can always be brighter, a voice purer, a mind sharper, a body stronger, a life longer. Everyone seeks to give their child the best chance but the most skilled geneticists are only accessible to a privileged few" [23] (p. 36). Gregor Wolbring has famously noted that the biotechnological efforts that aim to benefit the disabled are in fact far more concerned with enhancing, and extending, the boundaries of ableism. Wolbring sees this extension as a "transhumanization of ableism" that will "impact what we perceive as healthy bodies leading to the transhumanization of the meaning of health . . . . the scenario where only certain beyond species-typical body abilities are seen as healthy" [41] (p. 79). According to Humanity+ (formerly the World Transhumanist Association), transhumanism "affirms the possibility and desirability of fundamentally improving the human condition through applied reason, especially by developing and making widely available technologies to . . . greatly enhance human intellectual, physical, and psychological capacities" [42] (p. 68). In this case, Wolbring's extended model of ableism and subsequently, disability, creates a fundamental shift in what we perceive as "norms" today; enhanced bodies are the new norm whereas the unaltered become part of the disabled. Wolbring argues that:

> Disabled people using the latest assistive technologies, with their eyes fixed on medical progress, are a natural constituency for transhumanism. Disabled people in the wealthier industrialized countries, with their wheelchairs, prosthetic limbs, novel computing interfaces and portable computing, are the most technologically dependent humans ever known, and are aggressive in their insistence on their rights to be technologically assisted in fully participating in society . . . . Most disabled people think parents should have the freedom to choose to have non-disabled children and that technology can be used to overcome or cure disabilities, while we fight for equality for people with disabilities. Just as we should have the choice to get rid of a disability, we should also have the right to choose not to be "fixed," and to choose to live with bodies that aren't "normal." The right not to be coerced by society to adopt a "normal" body is also a central demand of transhumanism [43].

It is difficult to imagine parents rejecting genetic modifications that promise to improve the opportunities for their offspring; "the 'have nots'" and the "'haves'" will lead directly to an "ability divide" that is not only reflected on the individual, but sets a division on a global scale [37] (p. 254). In other words, Wolbring's new model of disability is decided by the access to technology and is very likely to "generate new ability divides as well as gradations of wealth from techno-poor to techno-rich" [43].

Genetic engineering has arguably been the largest and most lucrative industry of the last two decades, as medical, pharmaceutical, and production practices have capitalized upon biotechnological advance. Rosi Braidotti regards this commercial trend as an "opportunistic political economy of biogenetic capitalism" that turns "human and non-human intelligent matter—into a commodity for trade and profit . . . . The capitalization of living matter produces a new political economy . . . . Data banks of bio-genetic, neural and mediatic information about individuals are the true capital today" [37] (p. 61). This convergence between body, commodification and science is best articulated by Giovanni Berlinguer as "an absolutely novel situation compared to anything in the past," creating a situation that "derives from scientific advances which permit the removal, modification, transfer and use, for others' advantage (above all for health reasons, but not exclusively), of parts of the human body, of genes, of embryos" [44] (p. 35). Donna Haraway articulates the genomic commodification of the human body by stating "our authenticity is warranted by a database for the human genome . . . for the progress of science and the advancement of industry. This is Man the taxonomic type become Man the brand" [45] (p. 74). The human body's cellular material is not only manipulated but highly commodified, ensuring social hierarchy and maintaining a class structure that betrays the promise

of equal benefits technological advancement should bring. Consumer access to technology is very much a deciding factor in separating the *Gattaca* elite from the ostracized; Eugene's superior genetic stock reflects his family's wealth while Vincent's father has to trade in a vintage car for a genetically engineered second son. Class separation as a result of access, however, has always been a part of human society and seen in the inherently exclusive access to medical technology and pharmaceuticals today. With genetic engineering just around the corner, Wolbring believes that genomic commodification will soon become a "rats race for abilities . . . where the goal is to have the newest upgrades (abled), that one tries to out-able others by having better enhancements, that access to enhancements is seen as en-ablement and the lack of access as disenablement" [46]. As such, genetic manipulation is more than likely to further aggravate inequality; the rich and powerful would still be able to directly leverage economic inequality into intellectual and genetic inequality.

A growing body of scholarship focused on the rhetoric around disability and technology argues that technology inadequately serves disabled people. Alan Foley and Beth A. Ferri suggest that technology often reinforces ableism; they reassert the important argument that while technology increases access, it may also create "unexpected and under-critiqued forms of social exclusion for disabled people" [47] (p. 192). Disability rights advocates remind us that a technology which enables the selection of so-called "good" and "healthy" genes and "normal" traits will ultimately devalue the importance of diversity and disability as part of the human condition. Genetic editing will surely impact the social structure of disability, and cause even more discrimination to those already disabled. Braidotti stresses:

> There is a posthuman agreement that contemporary science and biotechnologies affect the very fibre and structure of the living and have altered dramatically our understanding of what counts as the basic frame of reference for the human today. Technological intervention upon all living matter creates a negative unity and mutual dependence among humans and other species. The Human Genome Project, for instance, unifies all the human species on the basis of a thorough grasp of our genetic structure. This point of consensus, however, generates diverging paths of enquiry. The Humanities continue to ask the question of the epistemological and political implications of the posthuman predicament for our understanding of the human subject. They also raise deep anxieties both about the moral status of the human and express the political desire to resist commercially owned and profit-minded abuses of the new genetic know-how [1] (p. 40).

The body's connectivity with science and technology continuously inspire discourse that rethink disability and the disabled's role in posthuman ways of being and becoming. Disability encourages a realignment of posthuman practices, challenging us to think again about the kinds of beings we value, accept, and include in the "accountable recomposition of a missing people" [1] (p. 100). Biotechnology negotiates the "alliances . . . made between humans and non-humans; 'between the organic and the inorganic, the born and the manufactured, flesh and metal, electronic circuits and organic nervous systems" [1] (p. 89). *Gattaca* provides a visual and textual lens much needed for a posthuman focus in disability studies, encapsulating the future of our social becoming while also exposing how biotechnology expands our species' genetic potential, with questionable consequences.

**Funding:** This research received no external funding.

**Conflicts of Interest:** The author declares no conflict of interest.

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
