# Peer review of "Perfecting Bodies: Who Are the Disabled in Andrew Niccol’s Gattaca?"

_philosophies, doi:10.3390/philosophies5020006_

Round 1

Reviewer 1 Report

This is an excellent article. I am convinced by the wide-ranging argument and, in particular, by your critique of previous readings of Gattaca. What the article lacks is an introductory paragraph - effectively a thesis statement - that sets out what you are going to explore, how, why and with what results in mind. (At present, you plunge in with Braidotti and it only becomes clear a couple of pages in just what the article is going to do.) Although you draw upon a wide array of sources, I would have liked to have seen some engagement with Gattaca as science fiction, especially as disability studies is an emerging area within sf criticism (cf. the pioneering work of Kathryn Allan or the online Journal of Science Fiction's recent disability issue). I did wonder whether, when referring to the social model of disability, you should have cited the original source - Mike Oliver - but that is a minor point. Two references (Jean Baudrillard and Francis Fukuyama) are missing from the Works Cited. 

Author Response

Dear Reviewer,

In this current revision, I've added an introduction that outlines the connection between posthumanism and SF texts. I've also elucidated the paradox between technology, cure, and disability in SF, drawing upon the research of Kathryn Allan, among others, to add more evidence to my claim. The close reading and textual analysis of disability in Gattaca are rearranged for a more cohesive discussion. All typos and citation notes have been addressed accordingly.

Thank you so much for your kind words and feedback!

Best Regards,

Celine Fahn

Reviewer 2 Report

Minor typos in the attached to be addressed. Otherwise, excellent.

Author Response

(The authors gave the same response as above.)
